# Mechanism Analysis of Nanosecond Pulse Laser Etching of SiC_p_/Mg Composites

**DOI:** 10.3390/ma15217654

**Published:** 2022-10-31

**Authors:** Zhe Wu, Jianyang Song, Yang Zhang, Bo Xue, Sijia Wang

**Affiliations:** 1College of Mechanical and Electrical Engineering, Northeast Forestry University, Harbin 150040, China; 2College of Science, Northeast Forestry University, Harbin 150040, China

**Keywords:** SiC_p_/Mg composites, laser etching, surface morphology, heat-affected zone

## Abstract

Due to the introduction of silicon carbide reinforcement, the physical and cutting properties of SiC_p_/Mg composites are very different from those of metal composites. Nanosecond pulse laser processing is more efficient than traditional processing for SiC_p_/Mg composites. A low-power pulsed fiber laser was used to etch 3.0 mm thick SiC_p_/Mg composites. The effect of low laser power (0~50 W) on the morphology and heat-affected zone of the SiC_p_/Mg composite after etching was studied. The results show that when the laser power increases, the material accumulation at the ablation end of the machining surface becomes more and more serious. With the increase in power, the differences in ablation width and ablation depth on the surface of composite materials do not increase proportionally. When the laser power increases gradually, the width of the heat-affected zone increases in the direction of the perpendicular laser beam and reaches the maximum value at the etched end.

## 1. Introduction

At present, cast magnesium alloy AZ91D and deformed magnesium alloy AZ31 are often used as substrates for magnesium matrix composites. The materials that can be used as reinforcers generally include ceramics, metals and intermetallic compounds. Ceramics are the most commonly used reinforcement materials in magnesium matrix composites, including carbides, borides and oxides. When the matrix is pure magnesium and magnesium alloy, silicon carbide particles are mostly used. Compared with other ceramic reinforcement materials, SiC has the characteristics of thermodynamic stability and better wettability for magnesium [1,2].

Magnesium matrix composites have excellent mechanical and physical properties. In the field of emerging technologies, SiC_p_/Mg composites have greater application potential than traditional metal materials and aluminum matrix composites. At present, magnesium matrix composites are mainly used in aerospace and military industries. In addition, SiC_p_/Mg composites have received international recognition in meeting environmental protection and sustainable development of alloy materials. With the new century, the importance of energy saving, environmental protection, recycling and other concepts is becoming more apparent [3]. Therefore, SiC_p_/Mg composites will have more applications in automotive and aerospace fields [4].

AZ91D magnesium alloy reinforced with silicon carbide particles was used in the experiment. The overall wear resistance, corrosion resistance and high temperature performance of magnesia-based silicon carbide composite material can be improved by silicon carbide particles [5,6]. At present, some material researchers have carried out studiees on the mechanical properties, physical properties and corrosion properties of magnesium matrix composites. The research on mechanical properties mainly focuses on tensile and compressive properties, fatigue and creep properties, wear resistance and so on. More research on the excellent wear resistance and corrosion resistance of SiC_p_/Mg composites has been carried out, compared to laser machining of SiC_p_/Mg composites. Susan et al. studied the abrasive wear behavior of SiC_p_/Mg composites. Corrosion resistance was studied by Mikucki et al., who carried out salt spray corrosion tests on SiC and believed that the corrosion rate remained unchanged below a certain critical value [6,7,8,9,10].

There are many difficulties involved in machining magnesium-based silicon carbide composites by traditional machining. Because of the high strength and hardness of magnesium-based silicon carbide composite material, the tools used are easily damaged in traditional processing. In the traditional machining process of SiC_p_/Mg composites, machining debris also easily damages the machining guide. Therefore, precision processing of magnesium-based silicon carbide composites is a major challenge in this field. Laser processing takes the laser as the processing energy, and the application and development potential of magnesium-based silicon carbide composite material processing are constantly expanding. It can realize non-contact machining and reduce the damage of the composite material caused by contact stress. The focused high-energy laser beam acts on the local region of the alloy with an energy value of more than 10^6^ T/cm^2^. In addition, the absorption rate of magnesia-based silicon carbide composite material with a long-wavelength laser is up to 80% [11,12,13,14,15], which can cause the material to melt and evaporate the instantly and achieve efficient processing. Because the focus spot is small, the heat-affected zone is small, which can meet the requirements of precision machining. The characteristics of low electromagnetic interference and easy guiding focus of the laser facilitate the realization of three-dimensional and special surface laser processing. So, laser processing technology is very suitable for processing SiC_p_/Mg composites [16,17,18].

In summary, there are few studies on the surface state and heat-affected zone of SiC_p_/Mg composites after laser machining. There is also a lack of sufficient experimental research on the micrometer morphology change in the SiC_p_/Mg composite surface. Therefore, it is necessary to study the surface state and heat-affected zone of SiC_p_/Mg composites after pulsed laser processing.

At present, the problem of precision and efficient machining of SiC_p_/Mg composites urgently needs to be solved. Laser processing can solve these problems, but research on laser processing of SiC_p_/Mg composites is still required. Therefore, this paper uses a low-power nanosecond pulse laser etching experiment and mechanism research on SiC_p_/Mg composite materials to solve the industry demand. At the same time, scanning electron microscopy (SEM) and ultra-depth of field (DOF) microscopy were used to observe and analyze the morphology and microstructure of the etched region under different machining powers. The effect of power on the laser etching effect of SiC_p_/Mg composites was studied. The results provide a theoretical basis for laser etching and laser cutting of SiC_p_/Mg composites [19,20].

## 2. Materials and Experimental Methods

### 2.1. Materials

AZ91D magnesium alloy composite with 20% mass fraction SiC particles was used in the experiment. The diameter of SiC particles ranges from 8 μm to 12 μm. The composite material was provided by Wang Xiaojun’s research group at the Harbin Institute of Technology, China. The rod-like material was obtained by powder stirring die casting [21]. The chemical composition of the AZ91D magnesium alloy is shown in Table 1. The physical properties of AZ91D and SiC are shown in Table 2. The sample is round, with a thickness of 3 mm and a diameter of 1 cm. The microstructure of the SiC_p_/Mg composites is shown in Figure 1.

It can be observed from the Figure 1 that the silicon carbide particles added into the SiC_p_/Mg composite material are about 10 μm in size. These bulk silicon carbide particles mixed with the AZ91D matrix can effectively prevent dislocation movement. This is also a more effective way to strengthen the metal matrix. The study of Kell et al. showed that the reinforcement effect of rod-shaped reinforcement with the same volume fraction was almost twice that of spherical reinforcement [22].

#### Material Handling

Before the experiment, rod-like raw materials are cut into circular slices by wire cutting, and then the sample is pressed into a cylindrical shape by a hot pressing machine after adding metallographic inlay powder, and finally, the surface is polished [23]. The test equipment adopts the pulse fiber multifunctional laser processing system. The experiment laser is a nanosecond pulse laser produced by China Jingwei Laser. The device model is JW-F30W, with a rated power of 50 W, laser wavelength of 1064 nm, repetition rate of 20–200 KHz, minimum linewidth of 0.01 mm, repetition accuracy of 0.0015, laser beam size of about 0.05 mm and a focal length of 19.5 cm. Due to the low-power etching, the temperature generated during processing is low, so the etching is carried out under the conditions of normal temperature and pressure, without adding auxiliary gas.

Before laser treatment, the surface of the SiC_p_/Mg composite substrate is gradually polished with 400#, 600#, 800# and 1000# metallurgical sandpaper to remove the oxide and oil on the surface, and cleaned with absolute ethanol, and dried naturally for use.

### 2.2. Etching Process Method

There are many process parameters for pulsed laser etching, and the average power of the pulsed laser is mainly selected as the variable in this paper [24]. We ensured that the pulse width, pulse frequency, discharge voltage, defocus, cutting speed and the other five process parameters remained unchanged. The laser power was varied so that the laser performed ten laser etchings on the surface of the experimental sample at different locations. By changing the value of the power parameter, the 3D scanning image, the real-time measurement image of the depth of the field and the SEM image are obtained.

The technological parameters of pulsed laser etching are shown in Table 3.

## 3. Theory

### 3.1. Pulse Laser Cutting Principle

The basic principle of pulsed laser cutting is to shine the pulsed laser onto the surface of the processed material. Under the action of the pulse laser, the processed material rapidly heats up due to the absorption of laser energy, burns, melts and evaporates, and forms free substances that are not attached to the matrix. At the same time, it is carried away by the coaxial jet air, which moves at a certain speed to complete the cutting process. Heating and removal are the key processes of laser cutting, and the relationship among the pulse energy, pulse frequency and cutting speed determines the cutting quality of the processed materials [25].

One can take the laser-melt cutting mechanism as an example. Figure 2a is a schematic diagram of the laser cutting process. The pulse energy determines the size of the melting region formed [26], namely the width w and height h, as shown in Figure 2b. Magnesium alloy sheet thickness is represented by *h*, laser frequency by *f*, and cutting speed by *v*. In order to complete the cutting process, the time required for the pulsed laser to apply to the thickness of the plate must be less than the time of taken to cover the walking *w* distance.
(1)$$t_{H}\leq t_{w}$$

In addition,
(2)$$t_{H}=\frac{H}{hf}$$
(3)$$t_{w}=\frac{w}{v}$$
(4)$$\frac{H}{hf}\leq \frac{w}{v}$$

It can, therefore, be concluded that
(5)$$v\leq \frac{hfw}{H}$$

Therefore, the melting point calculated by cutting speed, sample thickness and pulse frequency must meet Equation (5) before cutting the material. If evaporation and combustion are taken into account, the cutting speed can be even faster.

When the thickness of the cutting metal matrix composite plate is less than 2 mm, the slit is mainly formed by the sublimation of the material caused by the laser generation temperature, and the auxiliary gas cannot blow away the molten metal in time, and no hanging slag will be formed. When the thickness of the plate is greater than 2 mm, the area a third of the distance away from the upper surface is where gasification cutting takes place, and the slit below this is where melting cutting takes place. Combustion cutting depends on the oxygen content of the auxiliary gas [27].

### 3.2. Principle of High-Energy Beam Heat Treatment

Before establishing the solid heat transfer process, it is assumed that (1) the material is a one-dimensional semi-infinite solid; (2) the action power is constant; (3) cooling only depends on heat conduction; (4) the effect of latent τ heat on the temperature field can be ignored; (5) the thermophysical parameters do not change with temperature. By using the heat transfer model proposed by H.S. Kirslaw et al., namely the solid-state heat transfer equation that satisfies the classical Fourier heat conduction equation, the differential equation of laser beam heating can be given as follows:(6)$$cp\cdot \rho \cdot \frac{\partial T(x,t)}{\partial t}=\lambda \frac{\partial^{2}T(x,t)}{\partial x^{2}}+g(x,t)$$
where *ρ*—density of material; *λ*—thermal conductivity; *C_p_*—mass constant pressure and heat capacity; *α*—thermal diffusivity ($\alpha =\frac{\lambda }{\rho \cdot Cp}$); *g*(*x*, *t*)—the input heat flow expression, which can be further expressed as follows:
*g*(*x*, *t*) = *q*_0_·*δ*(*x*)·*H*(*τ* − *t*)

*q*_0_—The power density of the effective energy-carrying beam; *τ*—the total time of the energy beam interacting with the metal surface; *δ*(*x*)—the delat function; *H*(*τ* − *t*)—the Heaviside function.

This is obtained by integral deformation.

The heating process is as follows:(7)$$T_{h}(x,t)=\frac{q_{0}}{\lambda }\left[\sqrt{\frac{4at}{\pi }}\exp\left(-\frac{x^{2}}{4at}\right)-xerfc\left(\frac{x}{2\sqrt{at}}\right)\right]+T_{0}$$

The cooling process is as follows:(8)$$T_{c}(x,t)=T_{h}(x,t)-\frac{q_{0}}{\lambda }\left[\sqrt{\frac{4a\gamma }{\pi }}\exp\left(-\frac{x^{2}}{4a\gamma }\right)-xerfc\left(\frac{x}{2\sqrt{a\gamma }}\right)\right]+T_{0}$$
where erf(*x*)—the error function.
$$\mathrm{erf}(x)=\frac{2}{\sqrt{\pi }}\int_{0}^{x}e^{-u^{2}}du;$$

erfc(*x*)—The residual error function, where erfc(*x*) = 1 − erf(*x*); *T*_0_—the substrate temperature; *γ* = *t* − *τ*—the cooling time.

The relationship between energy beam power density and action time can be obtained as follows:(9)$$q_{0}\cdot t^{1/2}=\sqrt{\frac{\pi }{4\alpha }}\cdot \frac{\lambda (Tm-Ac)}{1-Ac/Tm}$$

A rule of practical significance in high-energy beam heat treatment is revealed, which is also applicable to SiC_p_/Mg composites etched by a pulsed laser. The energy beam power density *q*_0_ of high-energy beam heat treatment is inversely proportional to the square root of its action time t, which indicates that the larger the power density of high-energy beam action, the shallower the hardening depth [28,29,30,31,32,33].

## 4. Results and Discussion

When we use different powers to etch SiC_p_/Mg composites, the change in surface ablation gap can be observed by light focusing microscopy, as shown in Figure 3.

Figure 3 shows that the SiC_p_/Mg composites can be processed by gradually increasing the pulsed fiber laser power from the same ablation direction. When the power is increased from 5 W to 15 W, the surface of the composite material does not show severe ablation. At this time, the ablation depth is not obvious and the amount of slag hanging on the surface is less. When the power increases from 15 W to 20 W, a certain ablation depth appears at this time, represented by the blue areas in Figure 3. When the laser ablation moves inward along the set direction, the ablation depth decreases gradually. At this point, a small amount of material accumulation appears at the laser ablation end. When the power is increased to 25 W, the ablation depth is closer to the slagging at the end of laser machining. At this time, the hanging slag accumulation has not changed much compared with 20 W. When the power increases from 30 W to 50 W, the amount of hanging slag increases significantly compared with that of 25 W, and the ablation trajectory curve of the pulsed laser is more obvious. To sum up, the amount of slag hanging on the surface of the SiC_p_/Mg composite gradually increases with the increase in power. The surface roughness of the composite ablated after the power reaches 20 W is very high, and the increase in depth and width is obvious.

The changes in hanging slag and physical morphology on the surface of the laser-machined SiC_p_/Mg composites can be preliminarily observed from the real-time depth of the field map. The specific measurement chart below further verifies the surface changes in the laser-processed SiC_p_/Mg composites under different powers from the numerical simulation curve. A and B are the upper and lower limits of the measured depth, and C and D are the two limits of the measured width. This is shown in Figure 4.

According to Figure 4, the width and depth of the ablation zone on the surface of the SiC_p_/Mg composite after ablation can be obtained. When the power gradually increases, the numerical changes in the ablation width and depth can be observed, as shown in Figure 5.

It can be observed from Figure 5 that when the power reaches 5 W, the ablation depth is only 4.131 μm and the ablation width is 26.27 μm. When the power reaches 15 W, the ablation depth is about 14.24 μm and the ablation width is 54.58 μm, corresponding to the real-time depth of the field map. At this time, with the increase in laser power, the roughness of the machining surface decreases, and the machining depth and width are smaller. When the power reaches 20 W, the ablation depth is 18.20 μm and the ablation width is 62.19 μm. When the power is increased to 25 W, the ablation depth is 34.08 μm and the ablation width is 69.86 μm. When it increases to 50 W, the ablation depth is 79.15 μm, and the ablation width is 120.53 μm. At this time, the ablation width and depth have reached the maximum. The trend diagram shown in Figure 5c,d uses the differences in the width and depth values after the power changes.

It can be observed from Figure 6 that after the power of the pulsed fiber laser increases, the surface ablation depth of the SiC_p_/Mg composite increases more than the ablation width. Moreover, the ablation depth increases greatly when the power is increased to 20 W, which is larger than other power intervals. However, the ablation depth increases slowly when the power is between 30 W and 40 W. The increase in ablation width starts from 40 W. With the continuous improvement of laser machining power, both the ablation width and the ablation depth increase, but when the power is between 10 W and 15 W, the increase in the ablation depth difference is the smallest. When the power is between 40 W and 45 W, the increase in the ablation depth difference is the largest. When the power is increased between 5 W and 10 W, the difference in the ablation width is the smallest. When the power is between 20 W and 25 W, the ablation width difference is the largest. By comparison, it can be found that when the power is greater than 30 W, it mainly affects the ablation width, and when the power is less than 30 W, it mainly affects the ablation depth.. This is mainly because of the presence of silicon carbide in the composite. When the power increases, the melting rate of the magnesium alloy matrix is also faster. However, at the same time, the molten metal collects the unmelted silicon carbide at the bottom, preventing the laser from melting the underlying metal layer. So, the increase in the ablative depth difference will be smaller than the ablative depth difference.
(10)$$q_{0}\cdot t^{1/2}=\sqrt{\frac{\pi }{4\alpha }}\cdot \frac{\lambda (Tm-Ac)}{1-Ac/Tm}$$

By substituting the obtained parameters into Equation (9), it satisfies the law of practical significance in high-energy beam heat treatment. The higher the power density of high-energy beam, the shallower the hardening depth after etching.

After scanning electron microscopy (SEM) observation of the processed material, the SEM image of the composite material was obtained, as shown in Figure 7 below.

Figure 7e shows the typical observation results of the HAZ region during laser etching of the SiC_p_/Mg composites. When the power is 5 W~15 W, the state of the heat-affected zone on the surface of SiC_p_/Mg composites is not very obvious. Since the magnesium alloy with a lower melting point will be melted first and while SiC with a higher melting point is still on the surface of the original composite, silicon carbide particles will appear, as shown in Figure 7(a1). The width of the heat-affected zone in the upper half of the SiC_p_/Mg composite processing area is between 100 μm and 200 μm. A nanosecond pulsed laser etching device was chosen, so that a more pronounced frequency hole can be observed at the etched area (as shown in Figure 7(c2)) when the power reaches 15 W. Figure 7(d3) indicates that the surface is a smoother etched edge area after the power reaches 20W. At the end of laser etching, the width of the heat-affected zone evidently increases. Figure 7(e4) indicates the most severe example of oxidation of the material close to the etched area. After the power reaches 30 W, the edges of the etched area will start to form a raised hanging slag structure, as shown in Figure 7(f5). Therefore, the width of the heat-affected zone will gradually increase along the laser etching direction. After the power reaches 35 W, the inner wall of the hanging slag will form a more dense arrangement of silicon carbide wrapped by the magnesium alloy, as shown in Figure 7(g6). With the increase of power, the hanging slag will continue to accumulate and eventually fall off at the hanging slag edge, as shown in Figure 7(j7). The width of the heat-affected zone near the end of the laser etching procedure is obviously higher than that of the other regions. This is due to the rapid flow and oxidation of the molten matrix material at high temperature. Heat radiates to both sides of the processing area, and the presence of SiC particles makes the heat transfer speed faster.

Figure 8a,h show the elemental distribution of the EDS surface map analysis. The scan results show that the laser-etched SiC/Mg composites have a high content of Mg and O elements around the etched area, and many C and Si elements. This indicates that after laser processing, MgO and SiC particles accumulate on both sides of the etching zone. The distribution of oxygen elements can distinguish the distribution of the heat-affected zone and each area of SiC and AZ91D during laser etching of SiC/Mg composites.

When the laser processing power reaches 20 W, a large number of visible SiC particles are splashed out in the ablation process, as shown in Figure 9a. This is because the AZ91 matrix cannot combine with unmelted silicon carbide particles in the post-melting period. The silicon carbide particles are spattered out under the action of high-pressure gas and adhere to the master plate. At this time, the width of the complete rebinding zone and the incomplete binding zone around the etching region are 49.07 μm and 95.16 μm, and the width of the pure thermal shadow region is 107.53 μm. It can be observed using the scanning electron microscope that the probability of silicon carbide particles being spattered out in the ablation process increases under the laser processing power of 25 W. The width of the complete mixing zone is 154 μm, the width of the incomplete mixing zone is 265.93 μm, and the width of the partial fusion zone is 436.59 μm. The results show that when the power reaches 20 W, the silicon carbide particles will be splashed out of the processing area and will bind to the surrounding magnesium alloy matrix. This will affect the surface properties of SiC_p_/Mg composites and the heat transfer process.

As shown in Figure 10a,b, the silicon carbide particles bond to the AZ91D substrate under high power. In Figure 10a, it can be observed that a bonding region between the magnesium alloy substrate and the silicon carbide has been formed. In Figure 10b, the transition region between the recombined organization of silicon carbide and the magnesium alloy matrix and the original surface can be observed.

When the power is too high, the AZ91D magnesium alloy substrate will be melted by the laser. Due to the insufficient cooling time, it will bond with the silicon carbide particles and form a magnesium–silicon carbide recondensation structure, as shown in Figure 11c. When the laser beam passes through, the temperature drops from high to low. Due to the change in temperature, the recondensed structure will be damaged to some extent, as shown in Figure 11d. The presence of SiC particles directly affects the hardness, the depth of the hardened layer and the homogeneity of the tissue obtained after laser hardening of the SiC_p_/Mg composites. The presence of SiC particles leads to the inability of the laser to homogenize the microstructure under rapid heating and cooling conditions. Coarser granular carbides require a higher temperature and longer transformation time, which will directly affect the hardness and the depth of the hardened layer, and the microstructure will not be uniform. In conclusion, the finer the particles of the original microstructure, the finer and more uniform the microstructure of the laser-hardened layer will be under the conditions of rapid laser heating and cooling. Since the particles of silicon carbide chosen for the experiment are not coarse, a uniform laser-hardened layer can be obtained around the etched area after laser etching.

When SiC_p_/Mg composites are processed by the nanosecond pulse laser, the heat-affected zone diagram is shown as follows.

In the process of laser processing with low power and no auxiliary gas, the maximum power is 50 W. When the high-energy laser beam shines on the surface of the composite material, the surface temperature of the material rises rapidly in a very short time. When the temperature rises to the melting temperature and evaporation temperature of the composite, the magnesium alloy matrix and silicon carbide particles on the surface of the material are processed under the irradiation of the laser energy beam. When laser etching the surface of SiC/Mg composite material, the SiC particles with higher melting point will not melt, but pile up to the surrounding and sputter out silicon carbide particles, so that the etching zone changes. The mechanism is shown in Figure 12.

At this time, complex physical and chemical reactions occur on the surface of the composite, and the magnesium alloy matrix and silicon carbide particles change one after another. Shi et al. found that the interfacial reaction first formed nano MgO, and then whether it continued to react with liquid aluminum depended on the content of Mg and the density of the nano MgO layer [34]. Because the melting point of silicon carbide particles is higher than that of the magnesium alloy matrix, when the composite material melts into a liquid, the temperature keeps rising. The SiC_p_/Mg composite liquid phase first boils and then reaches the evaporation point. The metal vapor will cause the silicon carbide particles and melted matrix material to splash at high speed on both sides of the slit.

Because of the presence of SiC particles, the surface of SiC_p_/Mg composites after laser processing is different from that of ordinary magnesium alloys. When the high-energy laser beam is pointed at the magnesium alloy matrix without silicon carbide particles, the magnesium alloy matrix will rapidly evaporate and melt. When the magnesium alloy matrix melts and flows, it will immediately pile up on both sides. In this case, the content of the magnesium alloy matrix will change in the middle fully mixed region. However, when the high-energy laser beam first contacts the processed silicon carbide particles, the melting and evaporation rate of the magnesium alloy matrix decreases under the action of the silicon carbide particles. The microstructure of the silicon carbide particles coated by the magnesium alloy matrix and the sputtering phenomenon of silicon carbide particles can be observed.

More new bonded structures of silicon carbide and magnesium alloy will form in the composite materials in the partially mixed zone and incomplete mixed zone, which will improve the machining performance of the composite materials near the machining gap.

## 5. Conclusions

The surface mass and heat-affected zone distribution of SiC_p_/Mg composites etched by a pulsed fiber laser were studied. The main conclusions are as follows.
When SiC_p_/Mg composites are processed by low laser power, most of the molten matrix materials will be deposited at the end of the machining surface.When the laser power is increased, the etching depth of the composite surface increases gradually, and frequency holes, scum and microcracks can also be observed on the processed surface.With the increase in laser power, the width of the heat-affected zone increases along the beam direction, reaching a maximum value of 672 μm. The microstructure of silicon carbide particles wrapped by the magnesium alloy matrix is produced on the machining surface, and the sputtering phenomenon of silicon carbide particles occurs during the machining process, and the oxidation zone can be observed.With the increase in the laser power interval, the increase in the ablation width is the largest when the power is between 20 W and 30 W. When the power is between 40 W and 50 W, the variation in ablation depth is the largest.

## Figures and Tables

**Figure 1 materials-15-07654-f001:**
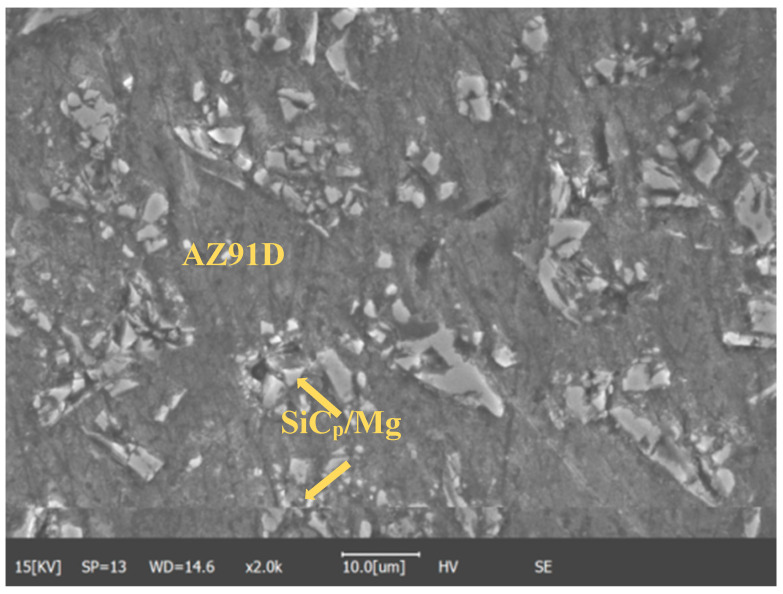
SEM of SiC_p_/Mg composite.

**Figure 2 materials-15-07654-f002:**
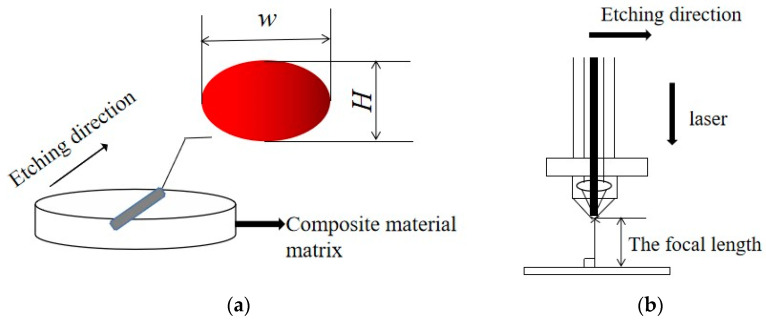
(**a**) Diagram of melting point; (**b**) cutting schematic diagram.

**Figure 3 materials-15-07654-f003:**
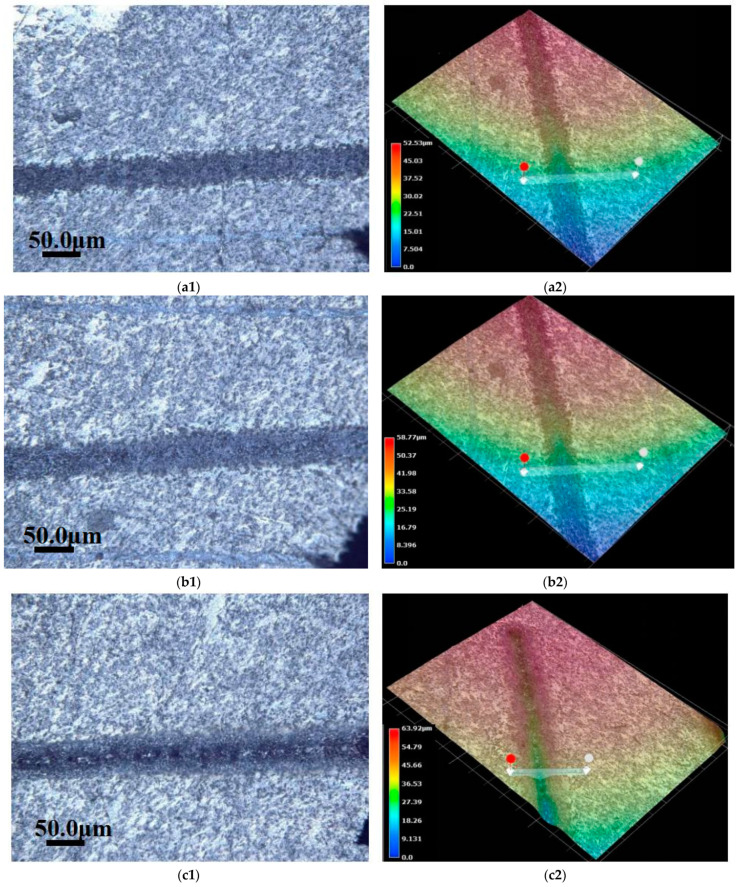
Laser confocal microscope real-time view and 3D view of the sample: (**a**) 5 W; (**b**) 10 W; (**c**) 15 W; (**d**) 20 W; (**e**) 25 W; (**f**) 30 W; (**g**) 35 W; (**h**) 40 W; (**i**) 45 W; (**j**) 50 W.

**Figure 4 materials-15-07654-f004:**
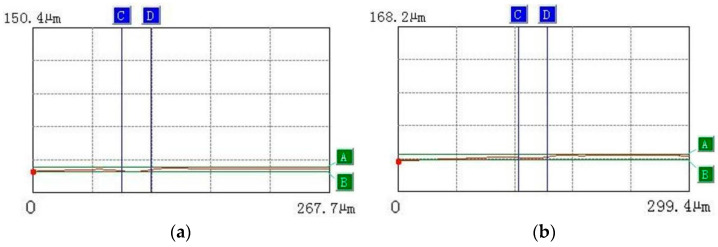
Numerical curve of light-focusing microscope: (**a**) 5 W; (**b**) 10 W; (**c**) 15 W; (**d**) 20 W; (**e**) 25 W; (**f**) 30 W; (**g**) 35 W; (**h**) 40 W; (**i**) 45 W; (**j**) 50 W.

**Figure 5 materials-15-07654-f005:**
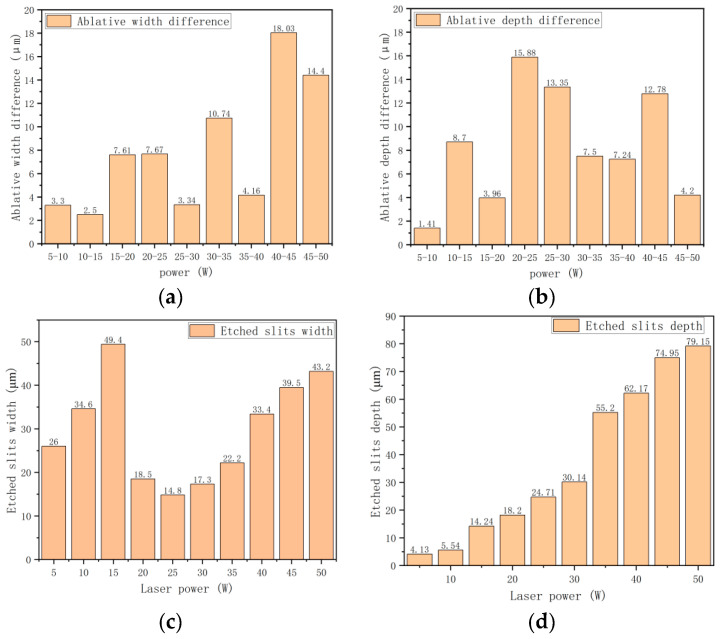
(**a**) Difference in the ablation depth; (**b**) difference in the ablation width; (**c**) ablation width trend diagram; (**d**) ablation depth trend diagram.

**Figure 6 materials-15-07654-f006:**
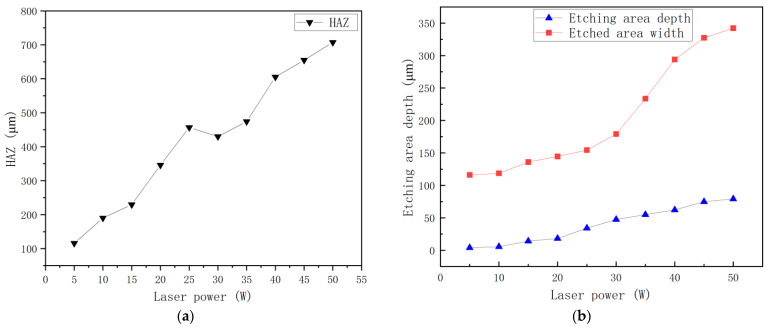
(**a**) Heat-affected zone trend graph; (**b**) comparison of etching width and depth variation.

**Figure 7 materials-15-07654-f007:**
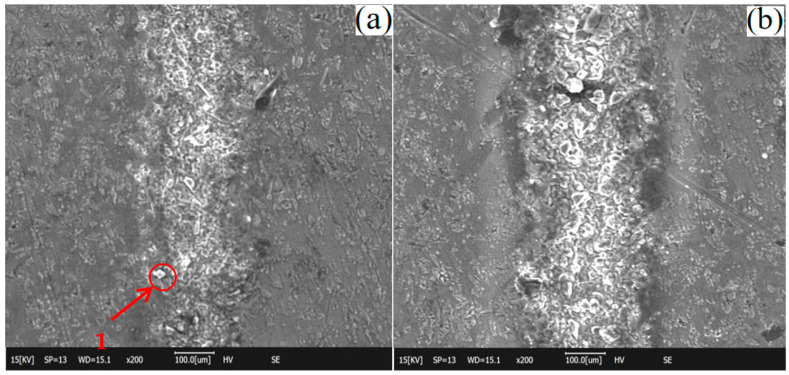
Scanning electron microscope (SEM) image. (**a**) 5 W; (**b**) 10 W; (**c**) 15 W; (**d**) 20 W; (**e**) 25 W; (**f**) 30 W; (**g**) 35 W; (**h**) 40 W; (**i**) 45 W; (**j**) 50 W.

**Figure 8 materials-15-07654-f008:**
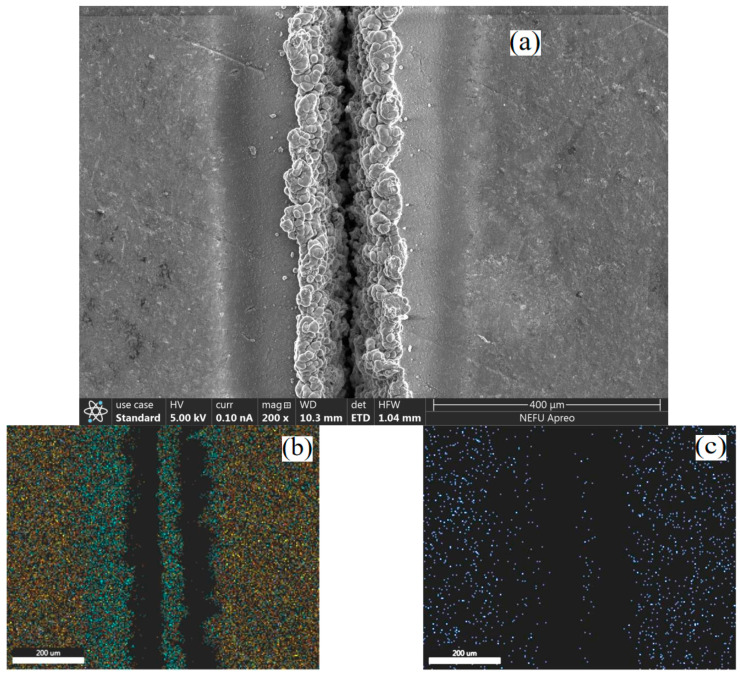
Laser etching of SiC/Mg composite microstructure and interfacial element distribution: (**a**) laser power 30 W, (**b**) general map of regional element distribution, (**c**) Al element distribution, (**d**) C element distribution, (**e**) Si element distribution, (**f**) Mg element distribution, (**g**) O element distribution; (**h**) peak content of each element.

**Figure 9 materials-15-07654-f009:**
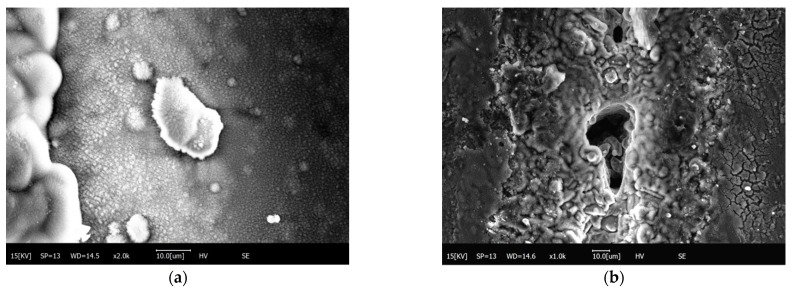
(**a**) Spatter of silicon carbide particles; (**b**) frequency hole.

**Figure 10 materials-15-07654-f010:**
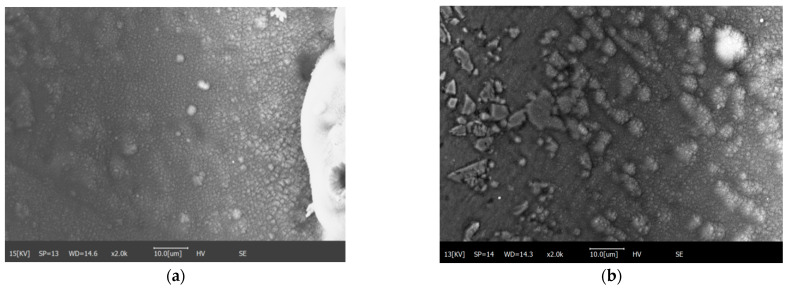
(**a**) Heat-affected zone transition area; (**b**) silicon carbide particles in the transition zone of heat-affected zone.

**Figure 11 materials-15-07654-f011:**
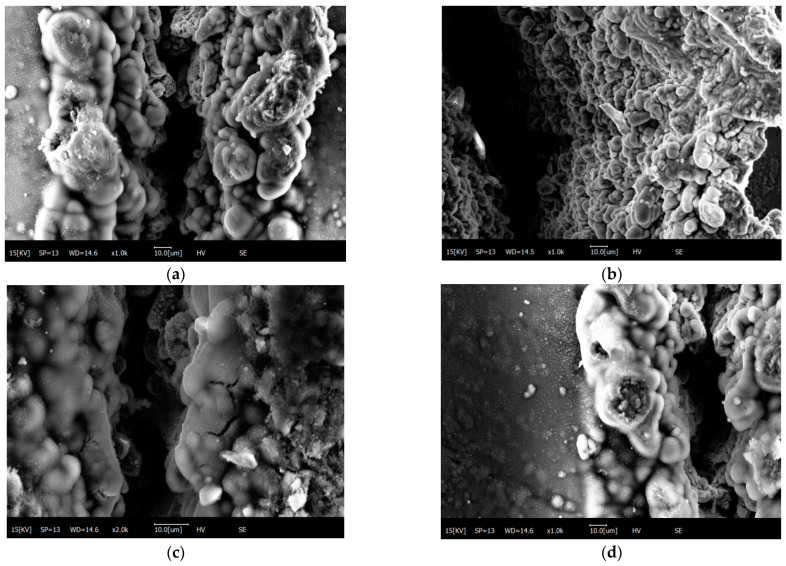
Bonding and fracture of sic particles with molten magnesium alloy. (**a**) SiC and matrix bonded structure; (**b**) The inner walls of the union are aligned; (**c**) Crack in combination; (**d**) Rupture of union.

**Figure 12 materials-15-07654-f012:**
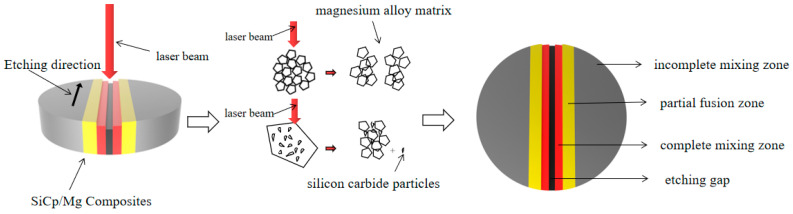
Schematic diagram of the formation of the impact zone of laser-etched SiC_p_/Mg composites.

**Table 1 materials-15-07654-t001:** Chemical composition of AZ91D magnesium alloy.

Alloys (wt%)	Al	Zn	Mn	Si	Fe	Cu	Ni	Mg
AZ91D	8.76	0.79	0.14	0.02	0.0003	0.001	0.008	Bal.

**Table 2 materials-15-07654-t002:** Physical and mechanical properties of AZ91D magnesium alloy and SiC.

Material	Melting Point	Thermal Conductivity	Modulus of Elasticity	Coefficient of Thermal Expansion
-	°C	W/(m·K)	GPa	10^−6^C^−1^
SiC	2300	83.6	450	4.7
AZ91D	596	72	45	2

**Table 3 materials-15-07654-t003:** Pulse laser etching parameters.

Samples	Pulsed Frequency *f*/KHZ	Laser Power /W	Scanning Speed *v*/mm·s^−1^	Laser Beam Diameter *D*/mm
1	20	5	20	0.01
2	20	10	20	0.01
3	20	15	20	0.01
4	20	20	20	0.01
5	20	25	20	0.01
6	20	30	20	0.01
7	20	35	20	0.01
8	20	40	20	0.01
9	20	45	20	0.01
10	20	50	20	0.01

## Data Availability

The data presented in this study are available from the corresponding authors upon reasonable request.

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
