# Peer review of "Mechanism Analysis of Nanosecond Pulse Laser Etching of SiCp/Mg Composites"

_materials, 2022, doi:10.3390/ma15217654_

Round 1

Reviewer 1 Report

1       Line 22, 25, 61, 96, 104, etc. - Please do not list References as a superscript.

Line 60 - 106T/cm2 =106 T/cm2

Before 2. Materials and Experimental Methods, please insert the objectives of this paper.

Line 75 – for 20%SiC powder, please provide SiC powder dimensional range introduced. In Fig. 1 is specified only the average 10um.

Please specify how many samples were tested for each power?

Table 1 – for the first column material = Material

Please specify laser beam dimensions, focus distance, defocusing?

Please specify the used gas? Gas flow?

In line 104 is cutting, in line 110 is etching (Table 3)?

Line 129, 130, 131 - In the formulas 1-1…1-5 is mentioned w and in Fig.2 is W.

Figure 2 - Incomplete drawing: (a) sample, cutting direction, (b) tilt angle, focus distance, cutting direction, gas flow

Line 136 - For metal matrix composite cutting, when the plate thickness is less than 2mm (missing of the predicate).

Line 154, 155, 156 – please see again the font size.

Line 170 - When different power is used to etch SiCp/Mg composites (missing of the predicate).

Fig. 3. – In line 80 is mentioned: The sample is circular, with a thickness of 3mm and a diameter of 1cm. In fig. 3 is another shape. Insert micrometers, not um, and everywhere um was used.

Fig 3. (f1) - This is ablation depth or ablation width?

Fig 3. (i1) - Recast is inside or outside the groove? Transversal section picture is missing.

Fig. 3. Sample light focusing microscope real view and 3D simulation - Explain the 3D simulation technique.

Table 4 – columns for sample = Sample and for power = Power

Table 4 - These values are obtained by measurement or by simulation. Please give details about the measurement technique.

Table 4 - last column/last value 79.15 µm - This in not cutting. Only 2.7% of the 3mm thickness was ablated at maximum tested power.

Fig.6. - What are the red arrows?

Line 252 – sic particles = SiC particles.

Line 276 – Please explain what means "too high", because the samples were not completely separated at maximum used power?

Line 283 – Please explain what is the coarse grain size?

Fig. 10 - Because the cutting was not complete, the ablation gap is not penetrated and complete mixing zone, partial fusion zone and incomplete mixing zone are around the ablated groove.

Line 304 - No information about these chemical reactions provided.

Please revise the writing of the references according to the indications.

Author Response

P.O. Box 3010, No. 24, He Xing Road

Harbin,Heilongjiang 150040, P. R. China

Tel: +86 451 82190397

      October  20, 2022

Dear Editors and Reviewers:

Thanks for your letter and for the reviewers' comments concerning our manuscript entitled“Mechanism Analysis of Nanosecond Pulse Laser Processing SiCp/Mg Composites”(ID: 1951305).Those comments are valuable for revising and improving our paper with important guiding significance. We have made correction according to the comments, revised portion are marked in red in the paper. The responds to the reviewer's comments are as follows:

Reviewer 1:

  1. Line 22, 25, 61, 96, 104, etc. - Please do not list References as a superscript.

Thank you for highlighting this deficiency. We have modified the citation format of references 22,25,37,41,49,62, etc.

  1. Line 60 - 106T/cm2 =106 T/cm2

Thank you for highlighting this deficiency. We have corrected the units on line 60.

  1. Before  Materials and Experimental Methods, please insert the objectives of this paper.

Thanks to the reviewer's suggestions. In order to be more clear and in line with the reviewer's concerns, the objectives of this paper is described as follows.

At present, the precision and efficient machining of SiCp/Mg composites is an urgent problem to be solved. Laser machining can solve these problems, but the research on laser machining of SiCp/Mg composites is still relatively few. Therefore, in this paper, low power nanosecond pulse laser etching experiment and mechanism study of SiCp/Mg composite materials are carried out to solve the industry demand. At the same time, scanning electron microscope (SEM) and deep of field microscope (DOF) were used to observe and analyze the morphology and microstructure of the etching zone under different machining power. The effect of power on laser etching effect of SiCp/Mg composites was studied. The results provide a theoretical basis for laser etching and laser cutting of SiCp/Mg composites.

  1. Line 75 – for 20%SiC powder, please provide SiC powder dimensional range introduced. In Fig. 1 is specified only the average 10um.

Thanks for the reviewer's suggestion. We have added in line 86 of the paper that the size range of the silicon carbide particles is from 8 to 12μm, with an average size of 10μm.

  1. Please specify how many samples were tested for each power?

Thank you for highlighting this deficiency. We have added the description of experimental samples in line 128. Ten sample regions, one region for each power etching, are etched ten times in total.

  1. Table 1 – for the first column material = Material

Thanks to the reviewer's suggestions. We have corrected the case of the word material. We are sorry for our negligence.

  1. Please specify laser beam dimensions, focus distance, defocusing?

Thanks for the reviewer's suggestion.We added the laser beam size of about 0.05mm and focal length of 19.5cm in line 117. The exact amount of scattering was not measured because the laser unit was optional.

  1. Please specify the used gas? Gas flow?

Thanks for the reviewer's suggestion.We have added a note about the auxiliary gas in line 118. Since the low power laser was chosen for etching, which generates a low temperature, the etching was done at room temperature and pressure, and no auxiliary gas was added.

  1. In line 104 is cutting, in line 110 is etching (Table 3)?

Thank you for highlighting this deficiency.We have standardized the processing method to etching throughout the text, and we apologize for the oversight in terminology.

  1. Line 129, 130, 131 - In the formulas 1-1…1-5 is mentioned w and in Fig.2 is W.

Thanks for the reviewer's suggestion.We have modified the laser travel distance w in Fig. 2.

  1. Figure 2 - Incomplete drawing: (a) sample, cutting direction, (b) tilt angle, focus distance, cutting direction, gas flow.

Thank you for highlighting this deficiency.We added the sample position and etching direction in Fig. 2a and the etching angle, laser focal length, and etching direction in Fig. 2b according to the reviewer's suggestion.

  1. Line 136 - For metal matrix composite cutting, when the plate thickness is less than 2mm (missing of the predicate).

Thank you for highlighting this deficiency.We have reworked this sentence in the text at line 166, and thank you again for your correction.

  1. Line 154, 155, 156 – please see again the font size.

Thanks to the reviewer's suggestions.We have made adjustments to the font for these three lines in the text.

  1. Line 170 - When different power is used to etch SiCp/Mg composites (missing of the predicate).

Thank you for your suggestion. We have revised the sentence in line 201 to read "When we use different power to etch the SiCp/Mg composites, the change of surface ablation gap can be observed by laser confocal microscopy, as shown in Fig. 3" Thank you again for your correction.

  1. 3. – In line 80 is mentioned: The sample is circular, with a thickness of 3mm and a diameter of 1cm. In fig. 3 is another shape. Insert micrometers, not um, and everywhere um was used.

Thanks to the reviewer's suggestions. The shape of the laser confocal map in Fig.3 is the intercepted surface from the machined surface etched area, so it appears different shape. Thank you again for pointing out the scale problem in Fig 3, and we have corrected the scale.

  1. Fig 3. (f1) - This is ablation depth or ablation width?

Thank you for highlighting this deficiency.The marked yellow font in Fig. 3(f1) indicates the ablation width, which has been revised, and we are very sorry for such an oversight.

  1. Fig 3. (i1) - Recast is inside or outside the groove? Transversal section picture is missing.

Thanks for the question raised by the reviewer.The recast layer in Fig. 3(i1) is in the interior of the groove and is the product of the magnesium alloy melting and recombining with silicon carbide. Since it is a low-power laser etching resulting in a too small etch slit, the cross section is temporarily unavailable under the original sample, so the plot of the cross section cannot be observed for the time being.

  1. 3. Sample light focusing microscope real view and 3D simulation - Explain the 3D simulation technique.

Thanks for the question raised by the reviewer. The 3D simulation here is generated by laser confocal microscopy measurements, which corresponds to the real-time plot on the left. I apologize for any confusion caused to you by my misrepresentation.

  1. Table 4 – columns for sample = Sample and for power = Power

Thank you for highlighting this deficiency. We have integrated the sample column and the power column in Table 4.

  1. Table 4 - These values are obtained by measurement or by simulation. Please give details about the measurement technique.

Thanks to the reviewer's suggestions.The etch width data in the data in Table 4 are measured by the tool in the SEM, and the etch depth is measured by the data plot generated by the laser confocal microscope in Fig. 4.

  1. Table 4 - last column/last value 79.15 µm - This in not cutting. Only 2.7% of the 3mm thickness was ablated at maximum tested power.

Thanks to the reviewer's suggestions. We have standardized the terminology. The ablation depth is relatively small because it is laser etching. The amount of material removed from the sample surface is also less with low power laser etching.

  1. 6. - What are the red arrows?

Thanks to the reviewer's suggestions.We have labeled all the red arrows in Fig. 6 and explained the meaning of the arrows separately in line 75 of the article. The specific meanings are as follows.

Since the magnesium alloy with lower melting point will be melted first, while the SiC with higher melting point is still on the surface of the original composite, silicon carbide particles will appear as shown in original Fig. 6(a)1.A nanosecond pulsed laser etching device was chosen, so that a more pronounced frequency hole can be observed at the etched area in original Fig. 6(c)2 when the power reaches 15W. Original Fig 6(d)3 indicates that the surface is a smoother etched edge area after the power reaches 20W. Original Fig. 6(e)4 indicates the most severe oxidation of the material close to the etched area. After the power reaches 30W, the edges of the etched area will start to form a raised hanging slag structure, as shown in original Fig. 6(f)5. After the power reaches 35W, the inner wall of the hanging slag will form a more dense arrangement of silicon carbide wrapped by magnesium alloy, as shown in original Fig. 6(g)6. After the power continues to increase, the hanging slag continues to accumulate so that the silicon carbide at the edges of the peeling, as shown in original Fig. 6(j)7.

  1. Line 252 – sic particles = SiC particles.

Thank you for highlighting this deficiency.We have made the case correction for SiC in line 305 of the text.

  1. Line 276 – Please explain what means "too high", because the samples were not completely separated at maximum used power?

Thanks for the question raised by the reviewer. Too high here refers to the amount of power in the low power processing range.

  1. Line 283 – Please explain what is the coarse grain size?

Thanks to the reviewer's suggestions.The reference here is to the presence of SiC particles that prevent the laser from homogenizing the microstructure under rapid heating and cooling conditions, rather than coarse grains. We apologize for the confusion caused by our wording error. We have made the correction in line 336.

  1. 10 - Because the cutting was not complete, the ablation gap is not penetrated and complete mixing zone, partial fusion zone and incomplete mixing zone are around the ablated groove.

Thank you for highlighting this deficiency. We modified the representation of the original Figure 10 to be more accurate, including etch direction, etch depth, and etch area. Since the etch trajectory is linear, these three regions are not positioned around the groove, but parallel to the etch direction.

  1. Line 304 - No information about these chemical reactions provided.

Thank you for highlighting this deficiency.We have added in line 360 of the original article the chemical reactions occurring here in laser etched SiC/Mg composites as well as the interfacial reactions. The literature was reviewed to understand the interfacial reaction process of high temperature oxidized SiC particles reinforced with Mg-Al alloy matrix composites and the thermochemical reaction occurring in AZ91D. Shi et al. found that the interfacial reaction first formed nano-MgO, and then whether the reaction continued with the aluminum solution depended on the Mg content and the degree of denseness of the nano-MgO layer.

Here are the references:

[1]Shi, Z,L.; Gu, M,Y.; Liu J, Y.; Liu G,Q.;et al. Interfacial reaction between oxidized silicon Carbide and aluminum-magnesium alloy . Chinese Science Bulletin.2001, 14, 1161-1165.

[2]Zhang H, Zhang L, Men G, Han N, Cui G. Ablated surface morphology evolution of SiCp/Al composites irradiated by a nanosecond laser. Surface and Coatings Technology. 2022, 429. https://doi. .org/10.1016/j.surfcoat.2021.127973.

  1. Please revise the writing of the references according to the indications.

Thank you for your suggestion.We have revised the format of the references and updated the references as required.

Special thanks for your comments.

We tried our best to improve the manuscript and made some changes marked in red in revised paper which will not influence the content and framework of the paper.We appreciate for Editors/Reviewer.

Yours sincerely,

Yang Zhang

Name: Yang Zhang

Reviewer 2 Report

The authors have done pulsed laser processing of SiCp/Mg composites. This manuscript has serious issues. I do have major concerns. 

1. The reviewer is not able to understand what is novel in this work. Please explain.

2. Only mechanism analysis cannot be accepted in a reputed journal like Materials. Value addition in terms of other studies such as numerical analysis or optimization or increasing the quantum of work reported or theoretical analysis is required. Mechanism alone can not be accepted. 

3. AZ91D magnesium alloy was prepared by adding 20%SiC powder. Why is this grade of Mg chosen and why only 20%SiC. Why not less or more?

4. What is the significance of this section: 3.2. Principle of High Energy Beam Heat Treatment?? There is nothing relevant to the results reported in the paper.

5. There will be more new bonded structures of silicon carbide and magnesium alloy in the composite materials in the partially mixed zone and incomplete mixed zone, which will improve the machining performance of the composite materials near the machining gap. What is the experimental proof for this claim?

6.When SiCp/Mg composites are processed by low laser power, most of the molten ma- 329 trix materials will be deposited at the end machining. Do the authors do machining or etching or processing??? It confuses the readers. Title reveals processing, methods reveal etching and conclusion is given as machining. Use single terminology.

7. What do the authors mean by the word tissue in the following? The original tissue has a direct effect on the hardness, the depth of the hardened layer and the uniformity of the tissue obtained after laser hardening.

8. The coarse grain size makes it impossible to realize the homogenization of the microstructure under the condition of rapid heating and cooling of laser. Thee grains are not at all visible in the microstructure. Authors need to go in for high magnification to prove the same.

9.  Microgranular carbides are easier to transform under high-energy laser beam, while flake carbides are more difficult to transform, but they transform faster than coarse granular carbides. There is no evidence of EDAX results to confirm the presence of carbides.

10. Authors need to cite recent and most relevant references such as https://doi.org/10.1016/j.optlastec.2022.108210, https://doi.org/10.1007/s13369-022-07256-9, https://doi.org/10.1016/j.surfcoat.2019.125249, etc.

11. There is no evidence of XRD to confirm the exact alloy composition and the reinforcement composition.   

12. What is the experimental evidence to distinguish HAZ, incomplete mixing, partial fusion and complete mixing. A SEM micrograph showing all these zones with EDAX is required to confirm the same. 

The authors need to address all the above points before the article is published. 

Author Response

P.O. Box 3010, No. 24, He Xing Road

Harbin,Heilongjiang 150040, P. R. China

Tel: +86 451 82190397

   October 20, 2022

Dear Editors and Reviewers:

Thanks for your letter and for the reviewers' comments concerning our manuscript entitled“Mechanism Analysis of Nanosecond Pulse Laser Processing SiCp/Mg Composites”(ID: 1951305).Those comments are valuable for revising and improving our paper with important guiding significance. We have made correction according to the comments, revised portion are marked in red in the paper. The responds to the reviewer's comments are as follows:

Reviewer 2:

  1. The reviewer is not able to understand what is novel in this work. Please explain.

Thank you for your suggestion.We have made additions to the text to add novelty to this paper. The specific additions are as follows.

SiCp/Mg composites are a new type of high performance material with excellent properties compared to traditional materials, such as high strength, high hardness, good wear resistance, light weight, corrosion resistance and long life. However, since it is difficult to process SiCp/Mg composites at present, laser processing is needed to solve the problem of difficult processing of SiCp/Mg composites.In this paper, laser etching experiments of SiCp/Mg composites were conducted using the controlled variable method to investigate the microscopic changes occurring during laser etching of SiCp/Mg composites and the changes in the heat-affected zone in the etched region by varying the laser power, which provides some theoretical references for the subsequent studies of laser-processed SiCp/Mg composites.

  1. Only mechanism analysis cannot be accepted in a reputed journal like Materials. Value addition in terms of other studies such as numerical analysis or optimization or increasing the quantum of work reported or theoretical analysis is required. Mechanism alone can not be accepted. 

Thanks for the question raised by the reviewer. We have added additional energy spectrum experiments and analyzed the reasons for the distribution of elements on the surface of the energy spectrum. The distribution of the heat affected zone was analyzed by EDAX results and compared with and our laser etching experimental data. In addition, we also performed numerical measurements based on the microscopic images of the scanning electron microscope and added numerical analysis of the heat affected zone at different power parameters. We added a comparison of the variation of surface etching depth and etching width after laser etching of SiCp/Mg composites. We have improved the diagram of the formation mechanism of the heat-affected zone so that the reader can understand the variation of the etched affected zone more clearly. We have provided additional descriptions of the chemical reactions occurring in laser etching and cited articles that provide the relevant theoretical basis.

  1. AZ91D magnesium alloy was prepared by adding 20%SiC powder. Why is this grade of Mg chosen and why only 20%SiC. Why not less or more?

Thanks for the question raised by the reviewer. AZ91D is more widely used and more cost-effective than AZ31B, AZ61A and other magnesium alloys, and there are more studies based on AZ91D, which has more theoretical research basis. So we choose AZ91D as the substrate. When SiC particles are added in amounts greater than 20%, it can cause inhomogeneity in the composite. It also causes SiC agglomeration during extrusion casting, which degrades the final composite properties. When the addition of SiC particles is less than 20%, the prepared composites have limited performance improvement. Therefore, the magnesium matrix composites reinforced with 20% SiC particles were finally selected.

  1. What is the significance of this section: 3.2. Principle of High Energy Beam Heat Treatment?? There is nothing relevant to the results reported in the paper.

Thanks for the question raised by the reviewer. The high-energy beam principle is applied here in line 66 of the article and the experimental results are analyzed to support the theoretical analysis of the article. And we planned the third part of the article to the second part of the experiments and experimental methods.

  1. There will be more new bonded structures of silicon carbide and magnesium alloy in the composite materials in the partially mixed zone and incomplete mixed zone, which will improve the machining performance of the composite materials near the machining gap. What is the experimental proof for this claim?

Thanks for the question raised by the reviewer. By scanning electron microscopy Fig. 7(a)(b), it can be observed that the closer to the etched region, the more SiC is accumulated. And it can be observed in the energy spectrum that there is more amount of C, Si, Mg, and O nearer to the etched region, which is more amount of MgO and SiC recombination. The organization is also more dense. The physical properties around the etched region are enhanced due to the high hardness and stability of SiC. We have reviewed the literature to learn about this type of related knowledge. The references are as follows.

  • Li,M.; Han, H.; Jiang, X.; Jiang, A feasibility study on high-efficient laser cutting of SiC particles reinforced aluminum matrix composite using single-pass strategy. Optik. 2022,265.https://doi.org/10.101 6/j.jleo.2022.1 69485.

Zhang H, Zhang L, Men G, Han N, Cui G. Ablated surface morphology evolution of SiCp/Al composites irradiated by a nanosecond laser. Surface and Coatings Technology. 2022, 429. https://doi. .org/10.1016/j.surfcoat.2021.127973.

  1. When SiCp/Mg composites are processed by low laser power, most of the molten ma- 329 trix materials will be deposited at the end machining. Do the authors do machining or etching or processing??? It confuses the readers. Title reveals processing, methods reveal etching and conclusion is given as machining. Use single terminology.

Thank you for your suggestion. This paper investigates the laser etching of SiCp/Mg composites and we have standardized the terminology throughout the paper.We have revised the title to "Mechanism Analysis of Nanosecond Pulse Laser Etching SiCp/Mg Composites" based on your suggestion. We apologize for any inconvenience caused by our oversight.

  1. What do the authors mean by the word tissue in the following? The original tissue has a direct effect on the hardness, the depth of the hardened layer and the uniformity of the tissue obtained after laser hardening.

Thanks for the question raised by the reviewer.The tissue product after laser processing here is a composite material of silicon carbide particles re-formed with liquid magnesium alloy. We apologize for the confusion caused by our misrepresentation.

  1. The coarse grain size makes it impossible to realize the homogenization of the microstructure under the condition of rapid heating and cooling of laser. Thee grains are not at all visible in the microstructure. Authors need to go in for high magnification to prove the same.

Thank you for highlighting this deficiency. Here it is not a grain but a silicon carbide particle, and some irregular silicon carbide particles make the structure formed after processing uneven.We apologize for the wrong wording here.

  1. Microgranular carbides are easier to transform under high-energy laser beam, while flake carbides are more difficult to transform, but they transform faster than coarse granular carbides. There is no evidence of EDAX results to confirm the presence of carbides.

Thanks for the question raised by the reviewer. We have added energy spectrum experiments to the paper and obtained the corresponding elemental distributions as shown in Fig 8(b). It can be seen from the distribution of elements therein that there are more SiC particles around the etched area.

  1. Authors need to cite recent and most relevant references such as https://doi.org/10.1016/j.optlastec.2022.108210, https://doi.org/10.1007/s13369-022-07256-9, https://doi.org/10.1016/j.surfcoat.2019.125249, etc.

Thank you for your suggestion.We have added citations to these references in the references based on your suggestions. These articles have been very helpful to us and we thank you for your suggestions.

  1. There is no evidence of XRD to confirm the exact alloy composition and the reinforcement composition.   

Thanks for the question raised by the reviewer.We have added EDAX energy spectroscopy experiments to obtain a map of the elemental distribution in the etched region of the composite. The presence of AZ91D and MgO around the etched area was determined, as well as the accumulation of more silicon carbide after the etched area was processed.

  1. What is the experimental evidence to distinguish HAZ, incomplete mixing, partial fusion and complete mixing. A SEM micrograph showing all these zones with EDAX is required to confirm the same. 

Thanks for the question raised by the reviewer.In order to be able to distinguish more clearly the distribution of the heat affected zone, we added EDAX energy spectroscopy experiments. The distribution of the heat affected zone is confirmed by the distribution of oxygen elements in it, as well as the different regions where the melted magnesium alloy binds to the silicon carbide particles. The added part is marked in red in line 297 of the text.

Special thanks for your comments.

We tried our best to improve the manuscript and made some changes marked in red in revised paper which will not influence the content and framework of the paper.We appreciate for Editors/Reviewer.

Yours sincerely,

Yang Zhang

Name: Yang Zhang

Round 2

Reviewer 1 Report

Thank you for your comments.

Reviewer 2 Report

The authors have incorporated the reviewers comments. I now recommend for acceptance.